# Isolation and Characterization of *Thermus thermophilus* Strain ET-1: An Extremely Thermophilic Bacterium with Extracellular Thermostable Proteolytic Activity Isolated from El Tatio Geothermal Field, Antofagasta, Chile

**DOI:** 10.3390/ijms241914512

**Published:** 2023-09-25

**Authors:** Bernardita Valenzuela, Francisco Solís-Cornejo, Rubén Araya, Pedro Zamorano

**Affiliations:** 1Laboratorio de Microorganismos Extremófilos, Instituto Antofagasta, Universidad de Antofagasta, Antofagasta 1240000, Chile; francisco.solis@uamail.cl; 2Instituto de Ciencias Naturales Alexander von Humboldt, Facultad de Ciencias del Mar y Recursos Biológicos, Universidad de Antofagasta, Antofagasta 1240000, Chile; ruben.araya@uantof.cl; 3Departamento Biomédico, Facultad de Ciencias de la Salud, Universidad de Antofagasta; Antofagasta 1240000, Chile

**Keywords:** *Thermus thermophilus*, El Tatio, geysers, proteases, thermozymes

## Abstract

The present study describes the isolation of an extremely thermophilic bacterium from El Tatio, a geyser field in the high planes of Northern Chile. The thermophile bacterium named *Thermus thermophilus* strain ET-1 showed 99% identity with *T. thermophilus* SGO.5JP 17-16 (GenBank accession No. CP002777) by 16S rDNA gene analysis. Morphologically, the cells were non-sporeforming Gram-negative rods that formed colonies with yellow pigmentation. This strain is able to proliferate between 55 and 80 °C with a pH range of 6–10, presenting an optimum growth rate at 80 °C and pH 8. The bacterium produces an extracellular protease activity. Characterization of this activity in a concentrated enzyme preparation revealed that extracellular protease had an optimal enzymatic activity at 80 °C at pH 10, a high thermostability with a half-life at 80 °C of 10 h, indicating that this enzyme can be classified as an alkaline protease. The proteolytic enzyme exhibits great stability towards chelators, divalent ions, organic solvents, and detergents. The enzyme was inhibited by phenylmethylsulfonyl fluoride (PMSF), implying that it was a serine protease. The high thermal and pH stability and the resistance to chelators/detergents suggest that the protease activity from this *T. thermophilus.* strain could be of interest in biotechnological applications.

## 1. Introduction

Among thermophiles, the genus *Thermus* has been frequently isolated in various ecosystems, such as terrestrial hydrothermal springs, marine hot springs, abyssal geothermal fields, hot water taps, compost piles, and rock surfaces [1,2,3,4,5]. The first isolation of a *Thermus aquaticus* strain [6] revealed the potential biotechnological applications of the thermostable enzymes (thermozymes) present in this genus [7,8]. These enzymes exhibited inherent thermal stability under the required conditions for their use. Notably, DNA polymerases have emerged as the most iconic example of such applications, with a significant impact on the field of molecular biology [9]. Currently, the genus *Thermus* comprises 28 species with validly published names listed in the List of Prokaryotic Names withstanding in Nomenclature (LPSN) (https://lpsn.dsmz.de/genus/thermus, (accessed on 16 May 2023)). *T. thermophilus*, one of the extensively studied species within this genus, possesses genomes characterized by their small size, with most of them below 2.5 Mbp [10]. Additionally, *T. thermophilus* demonstrates the ability to grow under both aerobic and anaerobic conditions. This bacterium has served as a valuable model for understanding thermophily within the genus. Like *T. aquaticus*, its enzymes have played a pivotal role in significant biotechnological advancements. Notable examples of thermozymes isolated from *T. thermophilus* include Tth polymerase [8], glucose isomerase [11], xylose isomerase [12], α-glucosidase [13], DNA ligase [14], and retrotranscriptase activity [15], among others.

Within the genus *Thermus*, multiple strains have exhibited extracellular proteolytic activity, showcasing their potential in industrial applications. For instance, aqualysin from *T. aquaticus* YT-1 displays optimal activity between 70 and 80 °C [16]. Caldolysin, a native protease from *T. caldophilus*, exhibited exceptional stability at 90 °C and resistance to detergents [17]. A protease isolated from *Thermus* sp. Rt41A demonstrates optimum temperatures between 70 and 90 °C [18]. The *Thermus* sp. Ok6 strain, isolated from hot pools in New Zealand, harbors a protease that exhibited optimal activity at 70 °C [19]. Additionally, a serine protease derived from *T. thermophilus* HB8, cloned and expressed in *Escherichia coli*, exhibited remarkable stability at high temperatures (75 °C) and operates effectively across a wide pH range of 5.0–9.5 [20].

Geothermal fields harbor unique and extreme environments that support the existence of life forms adapted to high temperature [21]. Consequently, these geothermal ecosystems serve as attractive sources for thermostable enzymes with potential industrial applications. El Tatio Geothermal Field (ETGF) is located in the high plateau of the Andes in Northern Chile (22°20′ S and 68° W) and situated at an elevation of 4200 to 4300 m above sea level (m.a.s.l.). It is the largest geyser field in the Southern Hemisphere and the third largest worldwide, after Yellowstone Park in the United States and Dolina Geiserov, Kamchatka, in Russia [22,23]. The ETGF boasts over 80 active geysers and more than 400 fumaroles, with boiling temperatures ranging between 86.2 and 86.4 °C [24]. It accounts for approximately eight percent of the world’s geysers, covering an area of approximately 10 km^2^ [22,25]. The ETGF holds significant ecological importance as a polyextreme habitat, characterized by high-altitude conditions (>4270 m above mean sea level), low precipitation rates (<100 mm/year), increased evaporation rates, intense UV radiation (PAR: 1206 μmol m^−2^, UV-A: 33.0 W m^−2^ and UV-B: 6.0 W m^−2^), high concentrations of heavy metals, lower surface boiling temperature, and large daily temperature fluctuations [23,26,27].

Several studies have evaluated the microbial diversity in ETGF, by microscopical analysis and DNA sequencing, identifying thermophilic bacteria, green bacteria, cyanobacteria, and diatoms [23,26,28,29,30]. There are few microorganisms that have been cultivated and isolated from ETGF, one example of which is the archaeon from a solfataric pool, *Methanofollis tationis*, first described as *Methanogenium tatii* [31,32].

The present study reports the isolation of a bacterium from the genus *Thermus*, named *T. thermophilus* strain ET-1, isolated axenically from a hot spring in the ETGF, located in the Chilean highlands. This bacterium exhibited extracellular proteolytic activity, along with high thermostability and unique biochemical properties, suggesting that it could be a valuable source of thermozymes with industrial and biotechnological applications.

## 2. Results

### 2.1. Isolation, Characterization, and Identification of CT 801 Strain

A thermophilic microorganism was axenically isolated from the thermal sludge taken from a fumarole containing a hot pool of water at 80 °C at the ETGF (Figure 1a). *T. thermophilus* strain ET-1 was able to proliferate at 80 °C in liquid media and at 70 °C in agar plates, which is the maximum temperature for agar plate cultures to prevent agar melting. Bacterial growth was observed after three to six days of incubation at 70 °C, resulting in orange colonies with a circular convex shape and 3–5 mm (Figure 1b). The Gram staining revealed Gram-negative bacilli cells without endospore formation (Figure 1c). The protease activity, tested on ATM agar plates supplemented with milk, exhibited halos around the colonies, evidencing extracellular proteolytic activity at 70 °C (Figure 1d).

The analysis of the 16S rRNA gene sequence identified this microorganism as a bacterium belonging to the *Thermus* genus. The sequencing data revealed that the *T. thermophilus* strain ET-1 shares a high similarity of 99% with *T. thermophilus* SGO.5JP 17-16 (762633), shown in Figure 2.

### 2.2. Effect of Temperature and pH on Growth

To further characterize this strain, we analyzed the growing properties of this microorganism in liquid TM media. The optimal growth temperature was 80 °C, reaching the stationary phase after 40 h of incubation (Figure 3a). The optimal pH growth was observed in the range between 7 and 9 with an optimum growth at pH 8 (Figure 3b) at 80 °C.

### 2.3. Determination of Proteolytic Activity in Native PAGE

Given the Native PAGE results of the supernatant partially concentrated protein from *T. thermophilus* strain ET-1, it can be seen that the highest proteolytic activity was detected in the fraction cut between 15 and 20 mm from the origin of migration (Figure 4c); this is consistent with the degradation halo in the zymography visible at the same migration distance (Figure 4a). In addition, it is possible to visualize a protein band at the previously mentioned migration distance (Figure 4b).

### 2.4. Effect of Temperature and pH on Protease Activity and Stability

The extracellular proteolytic activity was further characterized by its basic parameters and thermostability. The proteolytic activity shows an optimum temperature at 80 °C, with more than 50% of maximal activity observed between the temperatures of 60 °C to 90 °C; however, at 100 °C, the proteolytic activity decreased sharply to 10% (Figure 5A). Even though the pH effect showed more than 50% of the relative activity between a pH of 5–10, at pH 8 and 9 the activity was close to 100% (Figure 5B). The results of the thermoresistance profile indicated that the enzyme preserves nearly 10% of proteolytic activity after 24 h of incubation at 80 °C, displaying a half-life of 12 h at this temperature (Figure 5C). The heat stability of the protease activity was studied, exposing the enzyme to different temperatures from 26 to 100 °C for 1 h. Figure 5D shows that maximal enzymatic activity is maintained at different temperatures until 80 °C; however, at 90 °C the activity was reduced below 50%, while at 100 °C the enzyme is completely inactivated (Figure 5D).

### 2.5. Effect of Cations on Protease Activity and Stability

The effect of various cations at a concentration of 10 mM proteolytic activity was assessed (Table 1). Mn^+2^ and Fe^+2^ reduced the relative enzyme activity to 80 and 75%, respectively. However, K^+^ and Zn^+2^ inhibited the enzyme activity to 57 and 58%, respectively, compared to the control without cations.

### 2.6. Effect of Different Inhibitors on Proteolytic Activity

The effects of enzyme inhibitors on proteolytic activity were studied and are reported in Table 2. Proteolytic activity was slightly inhibited by the presence of EGTA (10 mM), EDTA (10 mM), Benzamidine-HCl (50 mM, Aprotinin (100 μg/mL), and leupeptin (50 µM). The latter was able to reduce the activity near to 25%, and most protease inhibitors inhibit the activity to 17%. However, the use of PMSF at 5 mM reduces the proteolytic activity to 34% (Table 2).

### 2.7. Effect of Denaturing Agents on Proteolytic Activity

The effects of denaturing agents are shown in Table 3. The surfactant agents Triton X-100 and Tween 80 and the denaturing agent HCl-Guanidine do not have any effects on the relative activity. The use of B-mercaptoethanol and H_2_O_2_ only reduced the activity by 10%. In contrast, the use of SDS and urea reduces the proteolytic relative activity to 84 and 75%, respectively.

### 2.8. Effect of Organic Solvent on Residual Activity

The effect of organic solvent (0.5 mM) on proteolytic stability is shown in Table 4. The relative activity after the incubation is not affected by chloroform, xylene, DMSO, hexane, and acetone at the concentration used. The presence of alcohols methanol, ethanol, and isoamyl alcohol decreases the activity to 74, 70, and 79%, respectively. However, the use of 2-propanol reduces the activity at 57%.

### 2.9. Effect of Commercially Used Detergents on Residual Activity

The proteolytic activity in the presence of liquid commercial detergents using two laundry detergents (D1, D2) and one dishwashing detergent (D3) at a final concentration of 1% (*v*/*v*) was studied. The residual activity showed that D1 and D3 reduce the proteolytic activity by about 10%, compared to D2 which reduces their residual activity by 24%, as shown in Table 5.

## 3. Discussion

Microorganisms demonstrate an extensive array of metabolic diversity, making them a common and easily accessible source of enzymes for industrial applications [33,34,35]. Among these microorganisms, thermophiles garner special attention owing to their exceedingly stable enzymes capable of withstanding high temperatures. This attribute positions them as promising contenders for diverse industrial operations. For instance, they excel in starch solubilization within the temperature range of 70–105 °C [36]. Furthermore, they prove invaluable in ethanol production derived from cellulose and hemicelluloses under elevated temperatures, thus streamlining distillation processes [37]. Additionally, in the realm of food processing, thermophilic enzymes play a pivotal role by catalyzing reactions at elevated temperatures, consequently boosting reaction rates [38], among other applications.

Geothermal fields stand as extreme environments, offering a distinctive ecological niche for life forms finely tuned to high temperatures [21,39,40]. These exceptional habitats represent a prime refuge for thermophilic bacteria, a fact substantiated by their discovery in various geothermal fields across the globe. Notable instances include the Manikaran hot springs in Himachal Pradesh, India [41], the geothermal sites of Hammamat Ma’in, Zara Dead Sea, Hammamat Afra, Al-Burbita, and Al-Hemma in Jordan [42], the Hot Springs of Erzurum, Turkey [43], the Grændalur and Hveragerdi regions in southwestern Iceland [44], Yellowstone National Park in the USA [6,45], the Deception Island volcano [46] in Antarctica, among numerous others sites. There are not many reports of axenically isolated bacteria or archaea from the ETGF in Chile. Among the exceptions is the report of Zabel et al. [31], in which a mesophilic methanogenic archaeon was isolated, and it was reclassified by Zellner et al. [32] as *Methanofollis tationis*.

In this study, we present the isolation of an extreme thermophilic bacterium from ETGF. Phylogenetic analysis of the isolated bacterium based on 16S rRNA gene sequences indicated that strain ET-1 is phylogenetically classified within the species *T. thermophilus,* a thermophilic bacterium first described in Japan [47]. This species corresponds to cells Gram-negative, non-sporulating, aerobic rods containing yellow pigment, presenting an optimum temperature for growth between 65 and 72 °C. However, this species is able to grow between 47 and 85 °C. *T. thermophilus* strain ET-1 shows similar characteristics, with a yellow pigment and similar thermal tolerances. In addition, *T. thermophilus* strain ET-1 is phylogenetically related to the species *T. thermophilus* JL-8, a bacterium isolated in the Great Boiling Spring (GBS) geothermal system, Great Basin, Gerkach, Nevada [48]. Although its genomes encode proteases and peptidases enzymes, an extracellular protease activity has not been reported, preventing a direct comparison with strain ET-1.

It Is known that enzymes synthesized by thermophile microorganisms are a key source of enzymes with outstanding high thermal stability [49]. For instance, within *T. aquaticus*, the endonuclease Taq I exhibits activity even at temperatures as high as 70 °C [50], while Taq I polymerase displays its optimal activity at 80 °C [51]. These enzymes, often referred to as “thermozymes,” commonly manifest peak activity levels at elevated temperatures, showcasing both thermostability and resistance to irreversible inactivation under high-temperature conditions, while remaining optimally active [52].

*T. thermophilus* strain ET-1 harbors an extracellular proteolytic activity distinguished by its exceptional stability at elevated temperatures, a characteristic not uncommon among proteases within the *Thermus* genus. For example, aqualysin I, a native extracellular alkaline serine protease by *T. aquaticus* YT-1, shows an optimal proteolytic activity at a pH of 10 and 70 °C [53]. The presence of Ca^2+^ further bolsters the enzyme’s stability, especially when subjected to heat, resulting in maximal proteolytic activity at 80 °C. Notably, aqualysin I maintain its stability when exposed to denaturing agents (7 M urea, 6 M guanidine HCl, and 1% SDS) at 23 °C for 24 h. Contrarily, the proteolytic activity of *T. thermophilus* strain ET-1 demonstrated an optimal activity at 80 °C and pH 10, in the absence of Ca^2+^. Despite these discrepancies, both proteases demonstrate exceptional thermostability and resilience against a variety of chemical agents. Within *T. thermophilus*, several thermoenzymes have undergone further characterization, revealing their optimal functionality within the temperature range of 50 to 80 °C [54,55]. Noteworthy are the enzymes of the protease category in this species, which have been cloned and expressed in *E. coli*, encompassing FtsH, an ATP-dependent metalloprotease with dual roles in membrane protein quality control and cell division [56], Lon protease, a homohexameric ATP-dependent protease highly conserved across species and involved in maintaining protein homeostasis [57], and Peptide deformylase, involved in the initial processing steps of nascent peptides and ensuring that proteins are properly folded and functional [58]. When these thermoactive enzymes are expressed within *E. coli*, a mesophilic host, they generally retain their thermal attributes, implying the inherent nature of these properties within the enzyme structure [52]. Although all these proteases—FtsH, Lon, and Peptide deformylase—from *T. Thermophilus* demonstrate activity above 70 °C and are designed to function optimally at high temperatures, they are not extracellular enzymes, and therefore they are crucial for intracellular processes, a distinctive feature when compared with the proteolytic strain ET-1.

The thermostability and thermoresistance attributed to the protease activity in *T. thermophilus* strain ET-1 align consistently with the characteristics of thermozymes isolated from similar microorganisms. For instance, in the case of *T. thermophilus* HB8 [59], a species valued as a source of thermozymes for biotechnological applications, its genome harbors the TTHA0724 gene coding for a subtilisin protease. Upon expression in *E. coli*, this protease demonstrated peak activity within the temperature range of 65 to 85 °C at pH 7.5. Moreover, it displayed remarkable thermal stability, retaining 50% of its activity even after 48 h at 75 °C [20], similar to strain ET-1. However, the optimal temperature and pH of the proteolytic activity of *T. thermophilus* strain ET-1 showed remarkable differences. While subtilisin protease from *T. thermophilus* HB8 exhibits a broad plateau of activity between 65 and 85 °C, strain ET-1 displays a sharp peak of activity at 80 °C. Similarly, significant differences are observed when the pH of proteolytic activity is analyzed.

Thermozymes exhibit not only exceptional thermostability but also heightened resilience against chemical agents in comparison to their mesophilic counterparts. These attributes render them exceedingly intriguing for industrial applications [8,60]. The proteolytic activity of *T. thermophilus* strain ET-1 remains unaffected by substances such as SDS and urea; akin traits have been observed in enzymes such as pepsin [61] and carboxypeptidase [62], both notably resistant to urea. Noteworthy is aqualysin I, which boasts an optimal pH at 10 and functions at 80 °C; it maintains its activity even in the presence of 7 M urea and 1% SDS [53]. Similarly, *Thermus* sp. strain Rt41A produces a protease that remains stable in the presence of 6 M urea and 1% SDS [63].

The proteolytic activity of *T. thermophilus* strain ET-1 was found to be inhibited by PMSF. This property is a hallmark of serine proteases, which comprise a serine residue forming a catalytic triad along with aspartic acid and histidine within the active site [63]. Similar serine proteases that function effectively at high pH include the subtilisin family (S8) proteases, a prominent subgroup of serine proteases [64]. These enzymes are initially synthesized as a precursor form known as prepro-subtilisin, encompassing a signal peptide responsible for protein secretion, a pro-peptide, and a mature domain [65,66,67,68]. Subtilisin proteases also exhibit notable resistance to wide variations in temperature and pH, often remaining undenatured by detergents and toxic metals [20,68,69]. Ongoing molecular characterization efforts are in progress to ascertain whether the extracellular proteolytic activity can be classified within this enzyme family.

The characterization of proteolyt”c ac’ivity reveals robust functionality across the range of pH values investigated (pH 5–10). Notably, this activity exhibits enhanced stability at elevated pH levels, a trait indicative of an alkaline protease [70]. More than 35% of the global industrial proteolytic enzyme production is attributed to alkaline proteases [71]. These enzymes have garnered significant global attention due to their diverse applications in biotechnology, spanning fields such as food processing, tanning, waste treatment, textiles, detergents, silver recovery from photographic plates, pharmaceuticals, and medical diagnosis [71,72,73,74]. Moreover, they exhibit exceptional activity and stability under challenging conditions involving temperature, pH, and the presence of oxidizing agents. This exceptional adaptability accounts for their versatility across various applications [75]. Alkaline proteases are notably harnessed in detergent formulations, with their optimal activity typically within a pH range of 9–12 [76]. Notably, the proteolytic activity demonstrated by the *T. thermophilus* strain ET-1 demonstrates a parallel array of qualities. This intriguing characteristic implies that this particular strain holds the potential to serve as a valuable source of thermozymes, with diverse applications across the biotechnological industry.

## 4. Materials and Methods

### 4.1. Sampling

Sludge samples were taken under aseptic conditions from the lower terrace of the ETGF. Measurements of pH and temperature in situ were carried out with the use of a Portable pH Meter (pH/Temp) (HI 99141, Hanna Instruments, Woonsocket, Rhode Island, USA). The samples were stored and transported at 4 °C and processed at the Laboratory of Extremophile Microorganisms at the University of Antofagasta.

### 4.2. Isolation and Culture

For the isolation of thermophilic aerobic bacteria, we utilized Natural Tatio Media (NTM), which consisted of a water sample from the El Tatio thermal spring (filtered through a 0.2 μm environmental water filter), supplemented with 0.25 g of yeast extract and 0.20 g of peptone (Oxoid Ltd.a., Basingstoke, UK). Subsequently, the medium underwent sterilization by autoclaving at 121 °C for 15 min. To create solid media, 2% *w*/*v* agar from (Oxoid Ltd.a., UK) was introduced to the solution and subjected to sterilization under the same conditions.

Following the isolation process, we formulated an Artificial Tatio Media (ATM) based on the chemical analysis of the water from El Tatio, reported by Fernandez Turiel et al. [28]. This medium consisted of a salt solution comprising the following elements: 416.21 mg/L of CaCl_2_, 8.22 mg/L of MgCl_2_, 234.5 mg/L of NaCl, 4697.4 mg/L of KCl, 16.763 mg/L of SiO_2_, 77.832 mg/L of NaHSO_4_, 14.377 mg/L of CsCl, 0.030 mg/L of FeSO_4_, 0.025 mg/L of CuSO_4_, and 0.025 mg/L of ZnSO_4_. This salt solution was supplemented with 0.25 g of yeast extract and 0.20 g of peptone from (Oxoid Ltd.a., UK).The medium was sterilized as indicated above, and, similarly, solid media (ATM agar) was used with 2% *w*/*v* agar (Oxoid Ltd.a., UK)

Two grams of thermal sludge samples were cultivated in 50 mL of NTM using bottles (capacity of 250 mL) at 80 °C under constant agitation for 4 days (250 rpm orbital shaker, (LabNet 311DS, Marshall Scientific, Co., Huntington, WV, USA). Subsequently, serial dilutions of the grown cultures were inoculated on ATM agar plates and incubated at 70 °C for 48 h in a humid chamber. Isolated colonies underwent a series of seven transfers on agar plates, all carried out under identical growth conditions, guaranteeing the persistence of colonies that consistently demonstrate the desired traits—such as uniform growth patterns and distinctive morphologies—across each transfer step. Subsequently, the obtained axenic cultures were meticulously preserved by storing them at −80 °C. This was accomplished by immersing the cultures in a medium broth containing 10% dimethyl sulfoxide (DMSO) and 50% glycerol. This storage method ensures the long-term viability of the cultures while mitigating potential damage caused by freezing and serves as a safeguard for the purity and integrity of the isolated thermophilic bacterial strains.

The colonies were routinely cultured in liquid ATM at 70 °C with agitation, as well as on agar plates in a humid chamber at the same temperature, for a period of 48 h. The colonies were characterized by their macroscopic characteristics (shape, elevation, color, and diameter of colony). Microscopic characteristics of the bacteria present in the colonies were evaluated using Gram’s staining, and images were captured using a Nikon OPTIPHOT-2 microscope (Tokyo, Japan). In addition, proteolytic activity of the colonies was screened using TM agar plates supplemented with 2% skim milk (*w*/*v*) at 70 °C.

### 4.3. Effect of Temperature and pH on Bacterial Growth

For determining the optimal growing temperature, 5 mL of medium was inoculated with 50,000 cells, and the proliferation rate was assessed at temperatures between 37 °C and 90 °C. Cell culture proliferation was measured by OD at 660 nm using a light spectrometer (Evolution 60, Thermo Fisher Scientific, Waltham, MA, USA) in triplicate samples. To determine the bacterial concentration for inoculation, cells from an overnight culture were washed twice in 1X PBS and harvested by centrifugation at 5000× *g* for 15 min; cells were finally suspended in 1X PBS and quantified by a DAPI staining (DAPI Counterstaining Protocol, ThermoFisher Scientific). To determine the optimal pH for bacterial growth, a similar protocol was used at different pH values (5–10) for 24 h at the optimal growth temperature of the bacteria (80 °C), using the following buffers: 50 mM of glycine-HCl buffer (pH 5.0), 50 mM of sodium acetate buffer (pH 6.0), 50 mM of phosphate buffer (pH 7.0), 50 mM of Tris-HCl buffer (pH 8.0), 50 mM of Na_2_HPO_4_-NaOH buffer (pH 9.0 and 10).

### 4.4. Phylogenetic Identification

Genomic DNA was extracted from axenic cultures in stationary phase using the Wizard Genomic kit (Promega, Madison, WI, USA) according to the manufacturer’s instructions. The 16S rRNA gene was amplified by polymerase chain reaction (PCR) using the Eubacterial primers 27F (5′-AGAGTTTGATCMTGGCTCAG-3′) and 1525R (5′-AAGGAGGTGWTCCAGCC-3′) [77]. The 16S rRNA was amplified under standard PCR conditions using a MJ Research PT-100 Thermal Cycler (MJ Research, Inc, Waltham, MA, USA). The PCR products were cloned in pGEM-T easy vector (Promega), and the positive clones were sequenced at the facilities of the Faculty of Biological Sciences of the Pontifical Catholic University of Chile. Nucleotide sequences were analyzed for similarity to the 16S rRNA gene using the BLAST program [77]. The sequence was also aligned with those of the reference strains using ClustalW [78]. The evolutionary history was inferred using the neighbor-joining method [79]. The percentage of the replicate trees in which the associated taxa clustered together in the bootstrap test (1000 replicates) is shown in the branches. The evolutionary distances were computed using the Maximum Composite Likelihood method [80]. The analysis involved sequences of 16S rRNA genes from 20 closely related strains, plus an outgroup sequence obtained from GenBank. All positions containing gaps and missing data were eliminated. There were a total of 1438 positions in the final dataset. Numbers at branch points refer to bootstrap percentages. The evolutionary analysis was conducted with MEGA software, version 7.0 [81].

### 4.5. Partial Concentration and Dialysis of Proteolytic Activity

The partial concentration of the proteolytic activity was performed by modifications of the method described by Singh and Bajaj [82]. Cultures of *T. thermophilus* strain ET-1 were cultivated in liquid medium in an incubator shaker at 70 °C for 48 h. The biomass produced was separated by centrifugation at 15,000 rpm for 30 min at 4 °C, and the activity in the supernatant was subjected to 80% ammonium sulfate precipitation and centrifugation at 15,000× *g* for 30 min to 4 °C. The pellet was finally suspended in 1 mL of 50 mM Tris-HCl buffer (pH 7.5) and dialyzed using SnakeSkin^TM^ Dialysis Tubing 10,000 MWCO (Thermo Fisher Scientific, Waltham, MA, USA) in 1 L of the same buffer, three times (3 L in total), to eliminate the salt. This enzymatic concentrate was stored at 4 °C and used for further enzyme characterization.

### 4.6. Proteolytic Activity Assays

The proteolytic activity of the isolated colonies was visualized on agar plates containing 5% skim milk (*v*/*v*) and 50 mM of Tris-HCl, pH 7.5 buffer and incubated at 70 °C for 24 h in a humid chamber. A halo of casein degradation around the hole confirmed proteolytic activity on the plate.

Proteolytic activity was also performed using casein as a substrate by modification of the Folin–Ciocalteu method [82,83]. One hundred microliters of the partially concentrated enzyme was mixed with 400 μL of 0.05 M Tris-HCl buffer (pH 7) containing 2% (*w*/*v*) casein and incubated at 80 °C for 60 min. The enzyme reaction was stopped by adding 1 mL of 10% (*w*/*v*) trichloroacetic acid. The mixture was incubated at 4 °C for 30 min, followed by centrifugation at 10,000 rpm at 4 °C for 30 min. Next, 500 μL of supernatant was mixed with 50 μL of NaOH and 500 μL of Na_2_CO_3_ solution (10%, *w*/*v*); finally, 50 μL of Folin–Ciocalteu reagent (1:1 diluted in distilled water) was added. The mixture was incubated for 10 min at room temperature, and the optical density was measured in a Vis/UV Evolution60 spectrophotometer (Thermo Scientific) at 580 nm. Protein concentration was estimated using bovine serum albumin as standard. Controls were carried out omitting the enzyme preparation and replacing them with a 100 μL volume of dialysis buffer and were subjected to the same incubations. After this treatment, the controls’ OD measurements were subtracted from the OD readings obtained from the samples.

### 4.7. Native PAGE

The partially concentrated protease activity was visualized in native PAGE gels. Fractionation of protein was carried out similarly to SDS-PAGE gels according to the Laemmli method [84] but omitting the SDS in its preparation. The proteins and activity fractionated in the native PAGE were detected in the following ways: 1. A gel lane was cut and placed on a 2% (*w*/*v*) agarose plate supplemented with 2% (*w*/*v*) casein buffered in Tris-HCl pH 7.5. The protease activity transferred by diffusion was visualized by incubation of casein plates for 2 h at 70 °C. After this incubation, the casein plates were stained with Coomassie Brilliant Blue R-250 staining solution (Figure 4a). 2. To visualize the proteins fractionated in the native PAGE, a gel lane was cut and stained using the Coomassie Brilliant Blue R-250 solution (Figure 4b). 3. A gel lane was cut and divided into segments of 0.5 and 1 cm in length. These gel segments were incubated with 500 µL of 2% (*w*/*v*) casein buffered in Tris-HCl pH 7.5 solution for 2 h at 70 °C. Finally, the caseinolytic activity was determined by the Lowry method (Figure 4c). All assays were performed in triplicate.

### 4.8. Effect of Temperature and pH on Protease Activity and Stability

In order to determine the optimal temperature, the proteolytic activity was carried out using casein as a substrate with incubations in the range of temperatures from 24 °C to 100 °C and revealed by the modified Folin–Ciocalteu method, carried out as indicated above. The maximal enzymatic activity was considered as 100% activity.

In the pursuit of determining the optimal pH, the protease activity was assayed over the pH range 5.0–10.0 at 80 °C for 60 min, using casein dissolved in the following buffer systems: 50 mM of glycine-HCl, pH (5.0), 50 mM of sodium acetate (pH 6.0), 50 mM of phosphate buffer (pH 7.0), 50 mM of Tris-HCl (pH 8.0), 50 mM of Na_2_HPO_4_⋅NaOH (pH 9.0 and 10).

In order to determine the thermoresistance at 80 °C, an enzyme preparation was incubated at this temperature for different time intervals (0–36 h), and the residual activity was assayed. The activity of the enzyme preparation at time = 0 was considered as 100% activity.

For the determination of the thermostability of the concentrated protease, the enzyme preparation was incubated at different temperatures between 24 and 100 °C for 60 min, and the residual activity was measured as indicated. An enzyme preparation not exposed to the temperature treatment was considered as 100% activity.

### 4.9. Stability of Protease Activity in the Presence of Cation, Inhibitor, and Other Chemical Reagents

The effects of mono and divalent cations (Na^+^, K^+^, Ca^2+^, Fe^2+^, Mn^2+^, Zn^2+^, Cu^2+^, Mg^2+^, Co^2+^) metal ions, at a concentration of 10 mM, on protease activity were investigated by adding them to the reaction mixture. The activity in the absence of any additives was taken as 100%. The effects of enzyme inhibitors on protease activity were studied using PMSF at a concentration of 5 mM, Aprotinin at a concentration of 100 μg/mL, Benzamidine-HCl at a concentration of 50 mM, Leupeptin at a concentration of 50 μM, ethylenediaminetetraacetic acid (EDTA), and ethyleneglycoltetraacetic acid (EGTA) at 10 mM. The crude enzyme was pre-incubated with each inhibitor for 30 min at 26 °C, and then the enzyme activity was estimated using casein as a substrate at pH 8 and 80 °C. The activity in the absence of inhibitors was taken as 100%. All the assays were performed in triplicate. The controls without the enzymatic extract used in these experiments were carried out in the presence of cations or inhibitors as indicated above. After determining the amino acid release by the Folin–Ciocalteu reagent, the OD measurements obtained in the controls were subtracted from those obtained in the samples containing the enzymatic extracts.

### 4.10. Effect of Denaturing Agents on the Proteolytic Activity

The stability of the proteolytic activity was studied with surfactants (Triton X-100, Tween 80, Dodecilsulphate (SDS), and oxidizing agent H_2_O_2_), all at a concentration of 1 mM. Reducing agents (Dithiothreitol (DTT, 10 mM) and 2-Mercapthoethanol (10 mM), and HCl-Guanidine (6M) and Urea (7M)) were studied by pre-incubating the crude enzyme for 30 min at 22 °C. The residual activity was measured at pH 8 and 80 °C. The activity of the enzyme without any additive was taken as 100%. All the assays were performed in triplicate. The controls without the enzymatic extract used in these experiments were carried out in the presence of the indicated reagents. After determining the amino acid release by the Folin–Ciocalteu reagent, the OD measurements obtained in the controls were subtracted from those obtained in the samples containing the enzymatic extracts.

### 4.11. Effect of Organic Solvent on Protease Stability

The crude extract was mixed with 20% (*v*/*v*) organic solvents such as methanol, ethanol, 2-propanol, isoamyl alcohol, hexane, chloroform, DMSO, acetone, and xylene at 4 °C for 16 h. The effect of organic solvent on protease activity was determined by residual activity under pH 8 and temperature 80 °C. The activity of the enzyme without any additive was taken as 100%. All the assays were performed in triplicate.

### 4.12. Stability and Activity in the Presence of Commercial Detergents

To check the compatibility of protease activity with commercial detergents, the protein extract was pre-incubated in the presence of three detergents (found available on the market in Chile). The activity was measured for 30 min at 80 °C and pH 10. The detergents were diluted in tap water to a final concentration of 1% *v*/*v* (to simulate washing conditions). The residual activity was measured at pH 7.5 and 80 °C. The activity of the enzyme without any additive was taken as 100%. All the assays were performed in triplicate.

## 5. Conclusions

This study describes the isolation of extreme thermophilic bacterium (*T. thermophilus* strain ET-1) from the ETGF in Northern Chile. These bacteria display a proteolytic activity in supernatants exhibiting broad activity across different pH ranges (pH 5–10) and temperatures (60–90 °C), with optimal activity observed at 80 °C and pH 10. Additionally, the proteolytic activity in *T. thermophilus* strain ET-1 demonstrates high stability under thermal conditions, retaining approximately 50% of its activity after 10 h of incubation at 80 °C. Furthermore, activity shows remarkable stability in the presence of denaturing agents, organic solvents, and ionic compounds. The inhibition of the proteolytic activity by PMSF strongly suggests the involvement of a serine protease in this activity.

These results suggest that the ETGF could be an important source of microorganisms and thermozymes, with potential biotechnological applications.

## Figures and Tables

**Figure 1 ijms-24-14512-f001:**
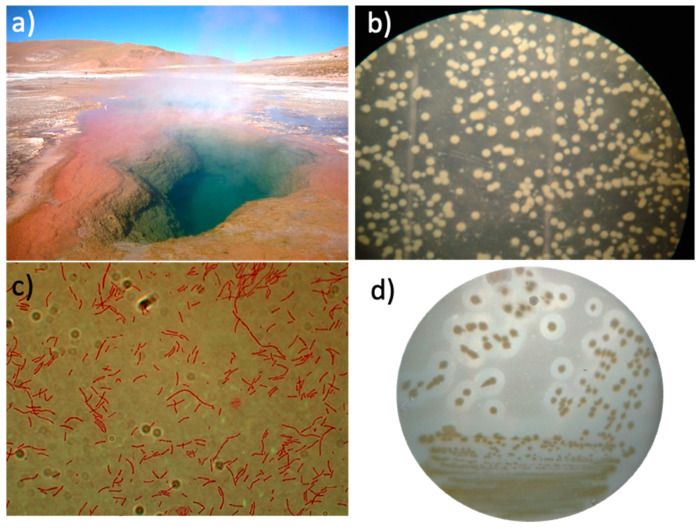
Isolation and characterization of *T. thermophilus* strain ET-1. (**a**) El Tatio Geothermal Field. (**b**) Axenic colonies isolated on 3% agar plates (*w*/*v*). (**c**) Gram staining of an individual colony (augmentation 1000×). (**d**) Halos of proteolytic activity of colonies on 3% ATM agar plates (*w*/*v*) supplemented with 2% skim milk.

**Figure 2 ijms-24-14512-f002:**
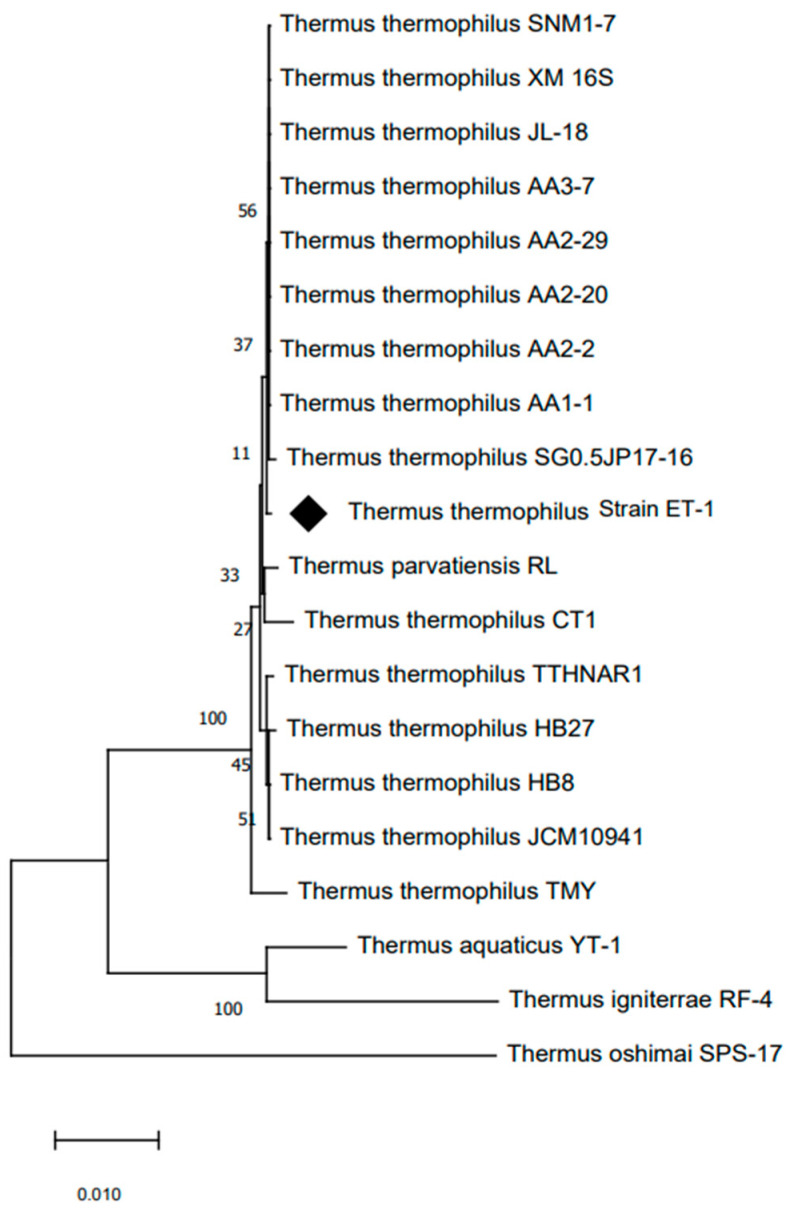
Neighbor-joining tree based on 16S rRNA gene sequences showing the phylogenetic relationship of *T. thermophilus* strain ET-1 (highlighted in black diamond). The numbers indicating the bootstrap values (percentage of 1000 replications) are shown at branch nodes. *Thermus oshimai* strain SPS-17 (GenBank: NR_026503.1) was used as an outgroup.

**Figure 3 ijms-24-14512-f003:**
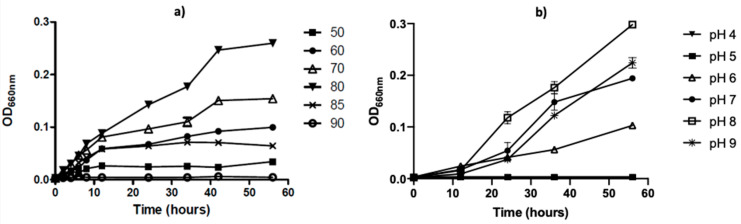
Temperature and pH effect on *T. thermophilus* strain ET-1 growth. (**a**) Comparative microbial growth on temperatures between 50 °C and 90 °C. (**b**) Comparative microbial growth on pH between 4 and 9 cultured at 80 °C. The results represent the average of triplicate samples ± SD. Error bars display standard deviations at each data point.

**Figure 4 ijms-24-14512-f004:**
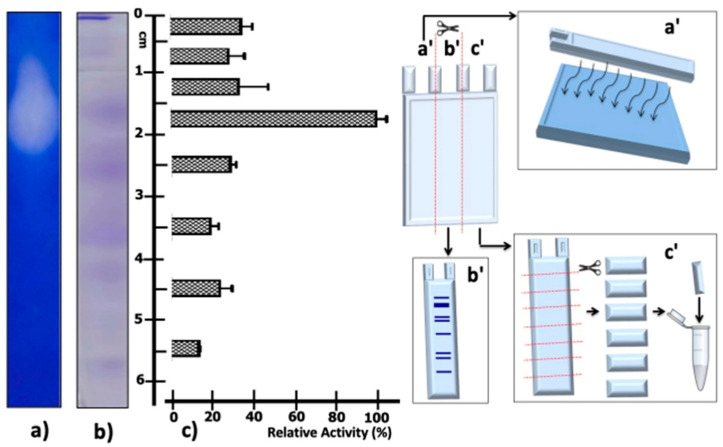
Native PAGE of the enzymatic extract from *T. thermophilus* strain ET-1. The native PAGE was processed as indicated on the left of the figure: Strips of the gel were excised to test for: (**a**) Caseinolytic activity zymogram, diagram of lane processing, and the post-revealed result of the halo of caseinolytic activity; this methodology is schematized in a′. (**b**) Native PAGE stained with Coomassie Blue; this methodology is schematized in b′. (**c**) Caseinolytic activity assays from sectioned native PAGE segments; relative caseinolytic activity is plotted on a horizontal bar graph. The results represent the average of triplicate samples ± SD. Error bars display standard deviations at each data point; this methodology is schematized in c′.

**Figure 5 ijms-24-14512-f005:**
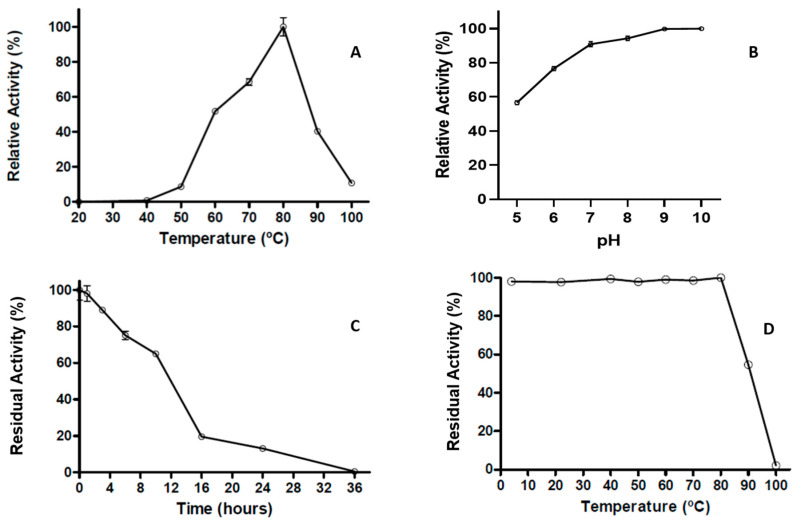
Optimal parameters of *T. thermophilus* strain ET-1 extracellular proteolytic activity. (**A**) Optimal temperature of proteolytic activity. (**B**) Optimal pH of proteolytic activity. (**C**) Thermoresistance of proteolytic activity to 80 °C (from 0 to 36 h). (**D**) Thermostability of proteolytic activity for one hour at different temperatures (from 4 to 100 °C). The results represent the average of triplicate samples ± SD. Error bars display standard deviations at each data point.

**Table 1 ijms-24-14512-t001:** Effect of cations on proteolytic activity.

Cation (10 mM)	Relative Activity (%)
Control	100
Ca^+2^	63 ± 4.2
K^+1^	57 ± 0.5
Mg^+2^	61 ± 3.2
Zn^+2^	58 ± 1.9
Mn^+2^	80 ± 10.8
Fe^+2^	75 ± 5.5
Cu^+2^	69 ± 6.6
Co^+2^	68 ± 4.6

**Table 2 ijms-24-14512-t002:** Effect of protease inhibitors on extracellular proteolytic activity.

Inhibitor	Concentration	Relative Activity (%)
Control	-	100
EDTA	10 mM	85 ± 3.7
EGTA	10 mM	93 ± 1.8
PMSF	5 mM	34 ± 1.6
Aprotinin	100 µg/mL	82 ± 3.8
Benzamidine-HCl	50 mM	83 ± 5.0
Leupeptin	50 µM	76 ± 5.0

**Table 3 ijms-24-14512-t003:** Effect of denaturing agents, detergent, and inhibitors on proteolytic activity.

Reagent	Concentration	Relative Activity (%)
Control	-	100
Urea	7 M	75 ± 1.0
HCl-Guanidine	6 M	103 ± 4.2
2-Mercaptoethanol	10 mM	90 ± 5.3
DTT *	10 mM	88 ± 5.5
H_2_O_2_	1 mM	90 ± 2.3
Triton X-100	1 mM	100 ± 4.4
Tween 80	1 mM	100 ± 3.0
SDS *	1 mM	84 ± 1.0

* DTT (dithiothreitol); SDS (sodium dodecyl sulfate).

**Table 4 ijms-24-14512-t004:** Effect of organic solvent on proteolytic stability.

Organic Solvent (0.5 mM)	Residual Activity (%)
Control	100
Methanol	74 ± 6.8
Ethanol	70 ± 9.2
2-propanol	57 ± 3.2
Isoamyl alcohol	79 ± 3.7
Hexane	96 ± 1.8
Chloroform	100 ± 11.2
Acetone	93 ± 0.3
DMSO *	98 ± 4.7
Xylene	102 ± 4.2

* DMSO (dimethyl sulfoxide).

**Table 5 ijms-24-14512-t005:** Stability and activity in the presence of commercial detergents (D).

Detergents 1% (*v*/*v*)	Residual Activity (%)
Control	100
D1 (OMO)	88 ± 2.4
D2 (ACE)	93 ± 3.0
D3 (QUIX)	94 ± 3.2

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
