# Peer review of "Isolation and Characterization of Thermus thermophilus Strain ET-1: An Extremely Thermophilic Bacterium with Extracellular Thermostable Proteolytic Activity Isolated from El Tatio Geothermal Field, Antofagasta, Chile"

_ijms, 2023, doi:10.3390/ijms241914512_

Round 1

Reviewer 1 Report

In the study the Authors presented an isolation of a new strain affiliated with Thermus genus, positive for an extracellular protease activity. Perhaps, it could add some new knowledge to a large known protease family, especially if the Authors manage to demonstrate how different the protease, they detected from others that already have been described. How beneficial its applications might be relative to other proteases?

My concern is mostly regarding “Method” part.  Proteolytic activity was measured by Lowry method. The Authors assumed that this assay would measure just amino acids and short peptides. Total Protein was assayed by Bradford method. Activity was expressed as “relative activity.” Relative activity must be defined.

Most important, Lowry method is used for total protein assay, as well as Bradford method.

Is it really possible for Lowry method distinguishing between short peptides and single amino acids vs. whole proteins?

How do we know that supernatant did not contain whole soluble proteins? I must note that centrifugation at 10.000g even for 30 min cannot precipitate soluble proteins. 

How was the method calibrated?

What were the controls to account for full length proteins that still might be in the reaction mix?

Another critical issue is that Lowry assay is non-compatible with various substances, including
TRIS-HCl 10 mM. In the study TRIS-HCl was used in the assay in 50 mM concentration. Also, the EDTA, EGTA should be used in the concentrations not higher than 1 mM. In the study they were used as 10 mM.  DTT is incompatible with Lowry. K+ and Mg2+ interfere with the assay. 

Before these concerns are addressed and revaluated, we cannot consider the data meaningful.

 I think the paper needs major re-writing.

I pinpointed some other issues that should be corrected or modified.

Abstract

L15-16. Re-write this sentence as “ morphologically the isolate was a non-sporulating rod-shaped bacterium. Colonies and Gram-staining go to the “Results”.

Line 19. Why was it compared to T. thermophilus strain HB8? Is it a type strain? If so, it should be indicated.

Intro

Line 52. Remove “Moreover”

Line 64 “States” plural

Line 76 modify as “…reported isolations of thermophilic bacteria from the ETGF”. Please, provide references on mentioned few studies.

“Methods”

Apart from the issues in methods indicated above, there are other comments here:

L 85-86. Sampling stated: “soil” was sampled. However, below I saw “sludge”. What was a sampling material: soil or sludge?

L. 86-87. T and pH were measured with a pH meter. Did pH-meter measured T too?

L. 91. Never start a sentence with “to” ( here and throughout the text). For instance, in this case:

“Thermophilic aerobic bacteria were isolated….”

Line 91-93. TM media description is not clear. Perhaps, you mean “consistent with water composition of El Tatio water”. Further you should present the composition of the medium in such way that any other laboratory can prepare it.

L. 94 not “by adding” but by addition”…

Is 2% agar enough to keep the medium solid at 70oC? Or was it semi-solid? Normally, for thermophilic incubations 4% is used. 

Line 90. The title of the sub-paragraph. Should be “ Isolation and cultivation”.

L. 98. Put “grown” before “culture”

L.99-101. The phrase is not well formulated. Perhaps, it’d better to modify: “isolated colonies were transferred seven times”

Between L.100 and L.101. Please insert information how the culture was routinely cultivated.

Sub-paragraph “Morphological characterization of the CT801 Strain”  is written not up to the standards, it is not enough and unacceptable. I suggest the Authors use Int. Journal of Systematic and Evolutionary Microbiology and look up any paper describing any novel strain how description should be done. 

L. 117-120. Has pHs were adjusted in accordance with cultivation at 70oC?

L. 136. The phrase “involved 11 nucleotide sequences” is not clear. Was a fragment length 11 nucleotides? Re-phrase, please.

L. 142. “homogenate” is not clear. Do you mean cell homogenate? Whole cells or crushed cells?

Homogenate in what?

L. 180. What was “untreated control” in the Effect of Temperature” and “Thermostability” experiments?

L. 187. “Stability of protease activity IN cation…” is bad wording. Use “Stability of proteolytic activity in presence of ….

Line 198. Effect ON , not in

Line 199. Use “with” instead of “in”

L. 223-224. Omit “this strain”. Start a sentence with “Strain -CT801”

L. 231. Analysis of 16S rRNA gene not confirmed but revealed or identified

L. 232. The findings could be from sequencing data, not from Figure 2. Modify this, please, as “Phylogenetic relationship of the 16S rRNA sequence….”

Figure 2. Indicate a type strain, please

Sub-paragraph “Partial purification…” is redundant to “Methods”. All details should be placed in “Methods”.

Figure 4c. What are the units for the activity? Should be marked on the graph.

Figure 5B. Optimal activity at the range of pH from 5 to 10. The graph should show parts with low or no activity too.

Line 295 Fe cation miss charge. It should be +2 or +3.

see attached above

Reviewer 2 Report

This manuscript by Bernardita et al. investigated a newly isolated thermophilic bacterium Thermus thermophilus obtained from a geyser field in the high-planes of Northern Chile. Morphological and phylogenetic analyses were performed for this thermophilic bacterium. The optimum growth conditions for this strain were also tested. The protease activity produced by this thermophilic strain was analyzed. This enzyme was also characterized for its optimum conditions and performance with different kinds of inhibitors.

This manuscript generally fits the scope of International Journal of Molecules Sciences. I have several comments which needs some revisions by the authors to the manuscript.

1.     Line 102, I think the temperature to store axenic cultures should be -80°C.

2.     Line 150, 159 and 160, the 10.000 g and 15.000 g should be 10,000 g and 15,000 g.

3.     Line 152, 0,5 M NaOH should be 0.5 M NaOH.

4.     Line 161, please add the method for dialysis.

5.     Line 187 to 197, please add the conditions used for enzymatic assay in this section.

6.     Please add the description about error bars in the legend of Figure 3 and 4.

7.     Line 254 to 266 was generally the description of methods used in this section. Please add some description about the results in Figure 4 for this section.

8.     In Table 1, 2, 3, 4, and 5, please add ± standard deviation from triplicate results to each residual activity percentage data.

9.     In Discussion, Line 341 to 346, please add more examples for thermophilic strains isolated from ETGF to support the statement in this paragraph.

10.  Line 376 to 378, please add reference to support this sentence.

Reviewer 3 Report

This manuscript reports on the isolation of a thermophilic bacterium from a geothermal field in northern Chile. The bacterium is identified as Thermus thermophilus and secretes proteolytic activity. The activity is characterized without either significant purification or molecular cloning. It shows an alkaline pH optimum, inhibition by PMSF and resistance to chelators and detergents. It is suggested that the unidentified protease responsible for the activity could be of interest for biotechnological applications. However, the study is rather preliminary for the reasons explained below.

MAJOR ISSUES

1.- The main problem of the manuscript, concerning its scientific content, is the lack of molecular identification and cloning of the enzyme responsible for the proteolytic activity. This precludes to evaluate the novelty of the finding. Is it a new protease or corresponds to other proteases already described? In lines 48-57, extracellular proteolytic activities observed in Thermus are mentioned and referenced. In lines 359-368, additional references for studies of Thermus proteases are given. In the manuscript, no explicit attempt is made to compare those proteases with the one described in this manuscript. In addition, in lines 394-395 it is stated that molecular characterization of the protease is underway. In summary, the current data are too preliminary.

2.- Concerning the formal presentation of the manuscript, a major problem is the sloppy reference list. It is totally non-professional: lots of references without journal source, article titles including terms that do not belong to the actual title, inconsistent formatting of journal names when mentioned… Too many mistakes to make a detailed list.

3.- Another formal defect of the presentation, is that the names of the authors are reversed: “Valenzuela Bernardita, Solís Francisco, Araya Rubén and Zamorano Pedro”, should actually be “Bernardita Valenzuela, Francisco Solís, Rubén Araya and Pedro Zamorano”.

4.- In lines 75-76, it is stated that “few studies have been reported for isolated cultures of thermophilic bacteria in the ETGF”. These “few studies” should be referenced.

5.- What criteria were applied, after the seven colony transfers mentioned in lines 99-102, to confirm the axenic character of the cultures obtained?

6.- The “partial purification” described in Methods (lines 157-162), is just a concentration step by ammonium sulfate precipitation. This is used to apply concentrated sample to electrophoretic gels for analysis. It is unclear whether the native PAGE is used as a purification step. It is also unclear what enzyme preparation was used for characterization in lines 274-331, Figure 5 and Tables 1-5.

7.- The data shown in Tables 1-5 do not give any evidence for the varibility of the results. Descriptions in Methods state that “All the assays were performed in triplicate”. Therefore all the data of Tables 1-5 should include estimations of variability. Relative activities should be given as means ± SD.

8.- There is some indefinition on the identification of the bacterium at the level of species. The title of the manuscript, the abstract (line 25) and the last paragraph of the Introduction (line 77) refer to it just by the genus name. However, in Figures 1-5, Discussion (line 348) and Conclusions (lines 408-409), it is described as Thermus termophilus. If this is OK, as it seems, the specific name should be used throughout, starting with the title.

9.- In lines 164-165 it is stated that the molecular weight of the protease was determined by SDS-PAGE. However, in the manuscript there is no mention to the result of this analysis. The results of the SDS-PAGE should be presented in a Figure and commented in the text.

MINOR ISSUES

In several cases, to express temperatures, the symbol ºC is used instead of °C which is the correct one.

In line 92, change “supplement” to “supplemented”.

In line 138, between “dataset” and “Numbers” a full stop is missing.

In line 152, change “released” to “release”

In line 163, change “proteases” to “protease”

In lines 192 and 303, and in Table 2 the use of “Aprotin” is mentioned. Shouldn’t this be “Aprotinin”?

The English is fine, only minor edits are required.

Round 2

Reviewer 1 Report

My comments are attached

All my comments are in the attached file

Reviewer 3 Report

The revised version of the manuscript is significantly improved. The main advance with respect to the first version is the change of focus from the biochemical characterization of the secreted protease, to the isolation, identification and characterization of the Thermus termophilus strain. This includes an adequate change of the manuscript title, with which I agree.

Anyhow, several points remain to be considered.

MAJOR POINT

1.- The authors still intend to describe the ammonium sulfate precipitation as a purification step (point 3 of their answer report, and paragraph on Partial purification and Dialysis in page 4). In support of their choice, they cite reference #38, another manuscript where ammonium sulfate fractionation is used as partial purification. I would like to bring to their attention the first paragraph of page 729 in that reference, where a progressive precipitation is implemented with increasing ammonium sulfate concentrations (not just a precipitation at 80% saturation). What is more important, in that manuscript, partial purification is confirmed by measurements of specific activity (units/mg protein).

The dialysis may or not achieve a marginal increase of specific activity, but primarily it is used to remove the excess of ammonium sulfate.

Therefore, to confirm that the 80% ammonium sulfate precipitation represents a partial purification of the protease, measurements of specific activity (units/mg protein) should be provided for the 15,000 rpm supernatant and the enzymatic extract after dialysis. The data should confirm an increase of specific activity.

MINOR ISSUES

In the final lines of the abstract Thermus sp. should be changed to T. termophilus.

Throughout the manuscript, the rule for microbial names should be full binomial name only at first mention in the abstract or in the text or in Figures and Tables. Thereafter the genus name should be abbreviated.

The expression “high thermal” is used in page 2 without being followed by a noun. For instance, it might be “high thermal conditions”, or substituted by “high temperature”.

In the Discussion, the term “specie” should be changed to “species” (this is the correct singular form of the word, valid also for the plural)

The reference list is clearly improved, but not yet correct. Most of journal titles are not properly abbreviated

The English is fine. Only minor edits are required.
